# Identification and Quantification of Urinary Microbial Phenolic Metabolites by HPLC-ESI-LTQ-Orbitrap-HRMS and Their Relationship with Dietary Polyphenols in Adolescents

**DOI:** 10.3390/antiox11061167

**Published:** 2022-06-14

**Authors:** Emily P. Laveriano-Santos, María Marhuenda-Muñoz, Anna Vallverdú-Queralt, Miriam Martínez-Huélamo, Anna Tresserra-Rimbau, Elefterios Miliarakis, Camila Arancibia-Riveros, Olga Jáuregui, Ana María Ruiz-León, Sara Castro-Baquero, Ramón Estruch, Patricia Bodega, Mercedes de Miguel, Amaya de Cos-Gandoy, Jesús Martínez-Gómez, Gloria Santos-Beneit, Juan M. Fernández-Alvira, Rodrigo Fernández-Jiménez, Rosa M. Lamuela-Raventós

**Affiliations:** 1Department of Nutrition, Food Science and Gastronomy, XIA, Faculty of Pharmacy and Food Sciences, Institute of Nutrition and Food Safety (INSA-UB), University of Barcelona, 08028 Barcelona, Spain; emily.laveriano@ub.edu (E.P.L.-S.); mmarhuendam@ub.edu (M.M.-M.); avallverdu@ub.edu (A.V.-Q.); mmartinezh@ub.edu (M.M.-H.); annatresserra@ub.edu (A.T.-R.); emiliami13@alumnes.ub.edu (E.M.); carancri77@alumnes.ub.edu (C.A.-R.); 2CIBER de Fisiopatología de la Obesidad y Nutrición, Instituto de Salud Carlos III (ISCIII), 28029 Madrid, Spain; amruiz@clinic.cat (A.M.R.-L.); sacastro@clinic.cat (S.C.-B.); restruch@clinic.cat (R.E.); 3Scientific and Technological Center of University of Barcelona (CCiTUB), 08028 Barcelona, Spain; ojauregui@ccit.ub.edu; 4Department of Internal Medicine, Institutd’Investigacions Biomèdiques August Pi i Sunyer (IDIBAPS), Hospital Clínic, 08036 Barcelona, Spain; 5Mediterranean Diet Foundation, 08021 Barcelona, Spain; 6Foundation for Science, Health and Education (SHE), 08008 Barcelona, Spain; pbodega@fundacionshe.org (P.B.); mdemiguel@fundacionshe.org (M.d.M.); adecos@fundacionshe.org (A.d.C.-G.); gsantos@fundacionshe.org (G.S.-B.); 7Centro Nacional de Investigaciones Cardiovasculares Carlos III (F.S.P.), 28029 Madrid, Spain; jesus.martinez@cnic.es (J.M.-G.); juanmiguel.fernandez@cnic.es (J.M.F.-A.); rodrigo.fernandez@cnic.es (R.F.-J.); 8The Zena and Michael A. Wiener Cardiovascular Institute, Icahn School of Medicine at Mount Sinai, New York, NY 10029, USA; 9CIBER de Enfermedades Cardiovasculares (CIBERCV), 28029 Madrid, Spain; 10Hospital Universitario Clínico San Carlos, 28040 Madrid, Spain

**Keywords:** polyphenol, phytochemical, biomarker, microbiota, dietary antioxidants

## Abstract

This study aimed to develop and validate a liquid chromatography/electrospray ionization-linear ion trap quadrupole-Orbitrap-high-resolution mass spectrometry (HPLC/ESI-LTQ-Orbitrap-HRMS) method to identify and quantify urinary microbial phenolic metabolites (MPM), as well as to explore the relationship between MPM and dietary (poly)phenols in Spanish adolescents. A total of 601 spot urine samples of adolescents aged 12.02 ± 0.41 years were analyzed. The quantitative method was validated for linearity, limit of detection, limit of quantification, recovery, intra- and inter-day accuracy and precision, as well as postpreparative stability according to the criteria established by the Association of Official Agricultural Chemists International. A total of 17 aglycones and 37 phase II MPM were identified and quantified in 601 spot urine samples. Phenolic acids were the most abundant urinary MPM, whereas stilbenes, hydroxytyrosol, and enterodiol were the least abundant. Urinary hydroxycoumarin acids (urolithins) were positively correlated with flavonoid and total (poly)phenol intake. An HPLC-ESI-LTQ-Orbitrap-HRMS method was developed and fully validated to quantify MPM. The new method was performed accurately and is suitable for MPM quantification in large epidemiological studies. Urinary lignans and urolithins are proposed as potential biomarkers of grain and nut intake in an adolescent population.

## 1. Introduction

The beneficial health effects of dietary (poly)phenols have been reported in several epidemiological and clinical trials [1,2,3], although their biological activities are not all attributed to their native form. After ingestion, modification by phase I and II metabolic enzymes reduces the concentrations of native (poly)phenols in the systemic circulation [4,5]. More than 80% of dietary (poly)phenols are not absorbed in the small intestine and reach the colon, where they undergo conjugation and are metabolized by gut microbiota through a range of enzymatic reactions (deglycosylation, dehydroxylation, demethylation, deconjugation, epimerization, ring fission, hydrolysis, and chain-shortening) [5,6,7]. The microbial phenolic metabolites (MPM) may be more bioactive than the parental (poly)phenol when they reach the target cells or tissues [8,9,10,11]. Fewer studies have reported MPM in young populations, such as adolescents.

High-resolution mass spectrometry (HRMS) using an Orbitrap mass analyzer is a well-established method for rapid targeted and untargeted identification of (poly)phenols in nutrimetabolomics studies [12]. This equipment provides exact mass information, two-stage mass analysis (MS/MS), and multi-stage mass analysis (MS^n^), which facilitates the structural elucidation of known and unknown compounds [12,13,14,15]. Therefore, HRMS constitutes a versatile and robust system for quantitative analysis [16,17,18,19]. However, to date, few methods are available to quantify MPM in human biological samples using high performance liquid chromatography (HPLC)/Orbitrap-HRMS.

The aim of this study was to develop and validate a high-performance liquid chromatography/electrospray ionization-linear ion trap quadrupole-Orbitrap-high-resolution mass spectrometry (HPLC/ESI-LTQ-Orbitrap-HRMS) method to identify and quantify urinary MPM in adolescents, and to explore the relationship of MPM with dietary (poly)phenols.

## 2. Materials and Methods

### 2.1. Study Design and Sample Selection

This work was carried out as a cross-sectional analysis within the SI! (Salud Integral) Program for Secondary Schools trial in Spain, a cluster-randomized controlled intervention trial (NCT03504059) aiming to evaluate the impact of a lifestyle educational program on cardiometabolic health in adolescents. A total 1326 participants were recruited in the baseline of the trial. Details of the study design, recruitment procedures, and Commission on Ethics are available elsewhere [20]. Informed consent was obtained for all the parents or caregivers.

For the current study, baseline data of 601 randomly chosen participants (53% girls) with available baseline urine samples were included, equivalent to 45% of the original cohort.

### 2.2. Chemicals and Urine Samples

The provenance of chemicals and standards is listed in the Appendix A. Urine samples were collected in 2017 and stored at −80 °C until analysis.

### 2.3. Sample Preparation and Extraction of (Poly)Phenols

All the spot urine samples were analyzed in a room with filtered light and kept on ice to avoid phenolic oxidation, following the procedure proposed by Martínez-Huelamo et. al., with some modifications [21]. Firstly, 1 mL of urine was acidified with 2 µL of formic acid and centrifuged at 15,000× *g* at 4 °C for 4 min. After centrifugation, the urine underwent a solid-phase extraction (SPE) and clean-up procedure using Waters Oasis HLB 96-well plates 30 µm (30 mg) (Waters Oasis, Milford, MA, USA). Plates were activated by consecutively adding 1 mL of methanol (MeOH) and 1 mL of 1.5 M formic acid. After loading 1 mL of sample, clean-up was performed with 0.5 mL of 1.5 M formic acid and 0.5% MeOH, and the elution with 1 mL MeOH acidified with 0.1% of formic acid.

The eluted fraction was evaporated to dryness under a stream of nitrogen gas in a sample concentrator (Techne, Duxford, Cambridge, UK) at room temperature, and reconstituted with 100 µL of 0.05% formic acid in water. The 96-well plate was then vortexed for 20 min and filtered through 0.22 µm polytetrafluoroethylene 96-well plate filters (Millipore, Burlington, MA, USA). To prepare calibration curves, synthetic urine was spiked with increasing concentrations of a mixture of 18 phenolic standards (3-hydroxybenzoic acid, 3-hydroxytyrosol, 3′-hydroxytyrosol-3′-glucuronide, protocatechuic acid, 4-hydroxybenzoic acid, vanillic acid, syringic acid, enterodiol, enterolactone, urolithin-B, gallic acid, dihydroresveratrol, urolithin-A, 3,4-dihydroxyphenylpropionic acid, 3′-hydroxyphenylacetic acid, *o*-coumaric acid, *m*-coumaric acid, and *p*-coumaric acid) before being processed and subjected to the same extraction procedure exactly as the samples. Abscisic acid d6 was used as an internal standard.

Synthetic urine was used as a blank, composed of calcium chloride (0.65 g/L), magnesium chloride (0.65 g/L), sodium chloride (4.6 g/L), sodium sulfate (2.3 g/L), sodium citrate (0.65 g/L), dihydrogen phosphate (2.8 g/L), potassium chloride (1.6 g/L), ammonium chloride (1.0 g/L), urea (25 g/L), and creatinine (1.1 g/L) [22].

### 2.4. HPLC/ESI-LTQ-Orbitrap-HRMS Instrumentation

#### 2.4.1. Chromatographic Conditions

Analysis was performed using an Accela chromatograph (Thermo Scientific, Hemel Hempstead, UK) equipped with a quaternary pump and a thermostated autosampler set at 4 °C, all operated by Chromeleon Xpress software. Chromatographic separation was accomplished with a reverse phase chromatographic column Kinetex F5 (50 × 4.6 mm i.d., 2.6 µm) (Phenomenex, Torrance, CA, USA) kept at 40 °C. Gradient elution was carried out with (A) water (0.05% formic acid) and (B) acetonitrile (0.05% formic acid) at a constant flow rate of 0.5 mL/min. The injection volume was 5 µL. A non-linear gradient was applied: 0 min, 2% B; 1 min, 2% B; 2.5 min, 8% B; 7 min, 20% B; 9 min, 30% B; 11 min, 50% B; 12 min, 70% B, 15 min, 100% B; 16 min, 100% B; 16.5 min, 2% B; 21.5 min, 2% B. The total run time was 21.5 min.

#### 2.4.2. Mass Spectrometry Parameters

Accurate mass measurements were performed on an LTQ Orbitrap Velos mass spectrometer (Thermo Scientific, Hemel Hempstead, UK) equipped with an ESI source working in negative mode. Mass spectra were acquired in profile mode with a setting of 30,000 resolution at *m*/*z* 400, and the mass range was from *m*/*z* 100 to 2000. Operation parameters were as follows: source voltage, 5 kV; sheath gas, 50 units; auxiliary gas, 20 units; sweep gas, 2 units, and capillary temperature, 375 °C.

### 2.5. Validation of the HPLC/ESI-LTQ-Orbitrap-HRMS Method

The method was validated following the criteria of the Association of Official Agricultural Chemists (AOAC) International in terms of linearity, limit of detection (LOD), limit of quantification (LOQ), recovery, intra- and inter-day accuracy and precision, and postpreparative stability [23]. All parameters were examined based on three concentrations (low, medium, and high) of each phenolic compound standard, as shown in Table 1.

#### 2.5.1. Linearity and Sensitivity

Calibration curves were prepared by spiking synthetic urine in triplicate using nine different concentrations of standard mixtures ranging from 1 to 1000 μg/L for 3-hydroxybenzoic acid, 3-hydroxytyrosol, 3′-hydroxytyrosol-3′-glucuronide, protocatechuic acid, 4-hydroxybenzoic acid, vanillic acid, syringic acid, *m*-coumaric acid, *p*-coumaric acid, *o*-coumaric acid, enterodiol, enterolactone, and urolithin-B; 2.5 to 2500 μg/L for gallic acid, dihydroresveratrol, and urolithin-A; 5 to 5000 μg/L for 3,4-dihydroxyphenylpropionic acid; and 10 to 10,000 μg/L for 3′-hydroxyphenylacetic acid, with the internal standard (IS) (+)cis, trans-abscisic acid d6 (500 μg/L). Calibration curves were created by quadratic regression analysis with residual plots lower than 15%. The adequacy of the model and linearity were assessed by coefficient of determination (R^2^).

LOD and LOQ were estimated for a signal-to-noise (S/N) ratio of 3 and 10, respectively.

#### 2.5.2. Accuracy and Precision

Accuracy was determined by analyzing five replicates of spiked synthetic urine with three known concentrations (Table 1) to evaluate the closeness of agreement between the calculated amount and the nominal amount of analyte. The results were expressed as the percentage of the ratio of the mean concentration observed and the known spiked concentration in the biological matrices. Precision was calculated using relative standard deviation (RSD) between the five spiked urine samples at three different levels on three different days. Intra- and inter-day precision was assessed using five determinations per three concentration levels (Table 1) in a single analytical run or on three different days, respectively.

#### 2.5.3. Recovery and Matrix Effect

Recovery and matrix effects (ME) were evaluated following the procedure described by Matuszewski et al., and Pereira-Caro et al. [19,24], analyzing three synthetic urines spiked at the three standard concentration levels (Table 1). Recoveries were calculated as the ratio between the area responses of standard concentration levels dissolved in pre-extracted samples and the analyte area responses of post-extracted urine spiked at the same concentrations. The results were expressed as recovery rate.

MEs were determined with the same concentration levels by comparing area responses of the spiked pre-extracted samples with the analyte area responses with neat standards dissolved in the mobile phase. The results were expressed as percentages. ME values above 100% are considered to indicate ion enhancement, and below 100% ion suppression.

#### 2.5.4. Stability

Postpreparative stability and freeze and thaw stability were assessed in this method. Postpreparative stability of the sample extraction process and during the time inside the autosampler at 4 °C were evaluated by injecting the post-extracted synthetic urine spiked with two standard concentrations (low and high) (Table 1) into the HPLC-ESI-LTQ-Orbitrap-HRMS system at 0 and 24 h. Freeze and thaw stability were assessed by injecting the post-extracted synthetic urine spiked at the same concentration levels into the HPLC-ESI-LTQ-Orbitrap-HRMS system after three freeze (−80 °C) and thaw (room temperature) cycles.

#### 2.5.5. Selectivity

The selectivity of the method was assessed by comparing chromatograms of blank human urine from three individuals and urine spiked with analytes at a known low concentration to discriminate between analytes and other endogenous components in urine.

### 2.6. Analysis of Urinary MPM by HPLC/ESI-LTQ-Orbitrap-HRMS in Adolescent Samples

#### 2.6.1. Targeted Identification of MPM

MPM were identified by comparing retention times with those of available standards. A semi-targeted screening method was established to identify phase II metabolites (glucuronides and sulfates) when reference standards were not available. The molecular formula of each compound was generated with an accurate mass and error of 5 ppm using the Xcalibur software v2.0.7 (Thermo Fisher Scientific, San Jose, CA, USA). Data acquisition techniques, including Fourier transform mass spectrometry (FTMS) mode (scan range from *m*/*z* 100–1000) in combination with product ion scan experiments (MS2) (Orbitrap resolution from 15,000 to 30,000 FWHM), were performed to obtain information about the *m*/*z* of precursor and fragment ions, retention time, and isotope pattern. Finally, analytes were confirmed by comparing MS/MS spectra with fragments found in the literature and The Human Metabolome Database 4.0 [25].

#### 2.6.2. Quantification of MPM

Calibration curves were constructed with available standards in synthetic urine and subjected to the same procedure as described above. To quantify phase II metabolites (glucuronides and sulfates), calibration curves of the aglycon form were used. Samples with concentrations that exceeded the highest point of the calibration curve were diluted and reinjected into the HPLC-FTMS system. Quantitative data processing was performed using Trace Finder software (LC version 4.1, Thermo Fisher Scientific, San Jose, CA, USA).

MPM concentration was normalized by urinary creatinine concentrations, which were determined using the Jaffé alkaline picrate method adapted to microtiter 96-well plates [26] and expressed as µg MPM/g creatinine.

### 2.7. Dietary (Poly)Phenols

Dietary intake was estimated using a semiquantitative food frequency questionnaire [27]. Dietary (poly)phenol intake was assessed by matching data from the Phenol-Explorer database v.3.6. [28]. Flavonoids, phenolic acids, stilbenes, lignans, phenolic acids, tyrosols, and other minor (poly)phenols, such as alkylphenols and alkylmethoxyphenols, were included in this analysis. Total (poly)phenol intake was estimated as the sum of individual (poly)phenol intakes and categorized into tertiles. Energy-adjusted (poly)phenol intake was calculated by the residual method established by Willet et al. [29].

### 2.8. Data Analysis

General characteristics of the studied population are presented as means (standard deviation (SD)) or median (interquartile range (IQR)) for quantitative variables and percentages (number) for categorical variables. MPM concentrations are presented as the mean, standard error of the mean (SEM). Student’s *t*-test was used to compare mean values of general characteristics between girls and boys, but also to compare MPM and postpreparative stability.

For the statistical analysis, MPM levels below the LOQ were set to values corresponding to half the LOQ. Pearson correlation was used to assess the relationship between urinary MPM and dietary (poly)phenols, as well as polyphenol-rich food sources. The false discovery rate (FDR) method was applied to adjust p-values for multiple correlations [30]. Data were normalized with the inverse normal distribution before the analysis.

The overall urinary MPM pattern and tertiles of total phenolic intake were assessed using principal component analysis (PCA) and presented as biplots in which eigenvectors were plotted as lines and the scores of individual samples as points. Beforehand, MPM data were standardized to unit variance.

Statistical analyses were conducted using the Stata statistical software package version 16.0 (StataCorp, College Station, TX, USA) and R v.4.1.1 (https://www.r-project.org, accessed on 1 April 2022). Statistical tests were two-sided, and p-values below 0.05 were considered significant.

## 3. Results and Discussion

### 3.1. Optimization of the HPLC/ESI-LTQ-Orbitrap-HRMS Method

Several SPEs solutions, as well as two SPE cartridges (Appendix A), were tested in order to obtain optimum recoveries. Two reverse-phase chromatographic columns were tested: Kinetex F5 (50 × 4.6 mm i.d., 2.6 µm) (Phenomenex, Torrance, CA, USA) and Atlantis T3 C18 (100 × 2.1 mm i.d., 3 µm) (Waters, Milford, MA, USA), obtaining better recoveries with Kinetex F5 and SPE 2 procedure (Appendix A). Different percentages of formic acid (from 0.05% to 0.1%) in mobile phases were tested to achieve desirable peak shapes and compound separation. The best results were obtained with 0.05% formic acid (data not shown). Two injection volumes (5 and 10 μL) were also tested to ensure optimum separation and detection of the analytes, and 5 μL sample injection gave the best results (data not shown). Details of the analytical conditions tested are available in the Appendix A.

### 3.2. Method Validation

#### 3.2.1. Linearity, LOD, and LOQ

The HPLC/ESI-LTQ-Orbitrap-HRMS method provided quadratic responses with coefficients of determination (R^2^) above 0.995 for all standards (Appendix A). Weighted factors (1/x statistical weight) were used to obtain the most reliable calibration curves.

The sensitivity of the method was evaluated by determining the LOD and LOQ of a synthetic urine sample spiked with standards. The LOD ranged from 0.02 to 3.29 µg/L, and LOQ from 0.06 to 10.96 µg/L.

#### 3.2.2. Precision and Accuracy

Intra- and inter-day precision varied in the ranges of 0−15% and 1−16%, respectively, in accordance with the values proposed by the AOAC (RDS < 15%) [23]. However, inter-day precision values for the lowest concentration of gallic acid, 3-hydroxytyrosol, and 3,4-dihydroxyphenylpropionic acid were 58, 31, and 26%, respectively (Appendix A), possibly due to early elution, which leads to a lower resolution peak when the concentration is low. Pereira et al. reported an intra-day precision of less than 15% (0% to 10%) for flavan-3-ols and their metabolites in a study using ultra high-performance liquid chromatography (UHPLC)-HRMS [19].

The accuracy was within the accepted limits of the AOAC guidelines [23] at all tested concentration levels for 89% of the metabolites analyzed, ranging from 80 to 120%. However, at the lowest concentrations the inter-day accuracy of gallic acid, 3,4-dihydroxyphenylpropionic acid, *m*-coumaric acid, and urolithin-A fell outside this range (Appendix A).

#### 3.2.3. Matrix Effect and Recovery

The average ME was 83%, with ranges from 53% to 126%, except those of urolithins-A and -B, which were below 35%. Minor ion suppression was also reported by Ordoñez et al. and Pereira-Caro et al. [18,19]. Ion enhancement was observed for 4-hydroxybenzoic acid and 3-hydroxyphenyacetic acid (Appendix A).

The average recovery of the three concentration levels was 89%, ranging between 70% and 99%. The lowest recovery was for gallic acid and 3-hydroxytyrosol, which was 70% at the lowest concentration (Appendix A). Similarly, Ordoñez et al. reported a mean recovery of 73% of urinary (poly)phenols extracted by an HLB cartridge and using an HPLC-HRMS method, obtaining a good recovery rate of 79% to 104% for free phenolic and glucuronide derivatives [18]. Better recoveries were reported by Pereira-Caro et al., with values ranging from 95% to 102% for 34 flavan-3-ol and its metabolites in rat urine samples analyzed by UHPLC-HRMS [19].

#### 3.2.4. Stability

The postpreparative stability assay showed no significant variation of analyte concentration in the urine matrix 24 h post-extraction at both low and high concentrations, except for 3-hydroxyphenylacetic acid, which was the analyte with the highest reduction (13%) (Appendix A). The freeze and thaw stability assay showed a signal decline of 14% for most analytes after the third freeze-thaw cycle. Likewise, Martínez-Huélamo et al., described a 12.9% reduction in signal for 3-hydroxyphenylacetic acid [21].

#### 3.2.5. Selectivity

Selectivity was confirmed by the absence of endogenous peaks in chromatograms at the same retention time as the analytes in three human urine samples. The method was, therefore, found to be selective for analytes at low concentrations and was able to discriminate between analytes and other components in urine.

### 3.3. Microbial Phenolic Metabolites Measured in Urine Samples

#### 3.3.1. General Characteristics of the Study Population

Out of the 601 randomized participants selected in this cross-sectional analysis, 546 had available information of food intake. The general characteristics of participants are presented in Table 2. The average age and body mass index (BMI) were 12.0 (0.4) years and 20.9 (4.2) kg/m^2^, respectively. The mean energy-adjusted (poly)phenol intake was 683.5 (335.3) mg/day. No differences were observed between boys and girls in terms of BMI and total (poly)phenol intake (Appendix A). Higher mean intakes of energy (*p*-value = 0.002), carbohydrates (*p*-value = 0.001), total fat (*p*-value = 0.010), and proteins (*p*-value < 0.001) were observed in boys (Appendix A).

#### 3.3.2. Identification and Quantification of Urinary MPM

Identification of MPM according to classes of (poly)phenols (lignans, hydroxybenzoic acids, hydroxycinnamic acids, hydroxyphenylacetic acids, hydroxyphenylpropanoic acids, stilbenes, hydroxycoumarins, and tyrosols) are presented in Appendix A. A total of 54 MPM were identified in urine. Enterolactone and urolithin diglucuronides were determined only in one sample.

Concentrations of MPM are summarized in Table 3. Excretion of urinary MPM varied highly between participants, and the majority of MPM were detected in the form of glucuronides and sulfates. Consistent with our results, Ordónez et al. reported that an HPLC-HRMS method was suitable for the analysis of phase II metabolites [18]. The most abundant MPM in the urine of all participants were phenolic acids, namely 3-hydroxyphenylacetic acid, hydroxyphenylacetic sulfate and glucuronide, protocatechuic acid sulfate-I, 3,4-dihydroxyphenylpropionic acid sulfate, hydroxybenzoic acid sulfate, and vanillic acid sulfate. These results are in agreement with Zamora-Ros et al., who detected phenolic acids as the most abundant urinary MPM in adult participants in the European Prospective Investigation into Cancer and Nutrition (EPIC) study [31]. Similarly, Hurtado-Barroso et al. found phenylacetic acids to be among the most abundant urinary MPM in young adults [32].

Urinary concentrations of stilbenes (dihydroresveratrol), tyrosols (3-hydroxytyrosol), and lignans (enterodiol) were low, with mean values below 10 µg/g of creatinine. Those reported by Zamora-Ros et al. in adults from the EPIC study were also low, being less than 5 µg/24 h [31]. These levels could be explained by a low dietary intake of stilbenes, tyrosols, and lignans, as reported in the food frequency questionnaire.

A high percentage of participants had a urinary MPM concentration below the LOQ for hydroxybenzoic acid glucuronide-I (38%), gallic acid (37%), 3-hydroxybenzoic acid (34%), enterolactone (30%), coumaric acid glucuronide-II (27%), 3-hydroxytyrosol (24%), and enterodiol (23%).

Interindividual variations in MPM could be explained by the gut microbiota profile, which is affected by age, gender, hormonal status, dietary habits, and other lifestyle variables [33]. In this study, the gut microbiota profile was not analyzed and thus the influence of the microbial family on MPM production was not determined.

Differences in urinary MPM between boys and girls are shown in Figure 1. Boys had higher values of 3,4-dihydroxyphenylpropionic, dihydroxyphenylpropionic sulfate, gallic acid, gallic acid sulfate, p-coumaric acid, vanillic acid glucuronide and sulfate, hydroxybenzoic acid glucuronide-I and sulfate-I, protocatechuic acid sulfate, 3-hydroxyphenylacetic acid and hydroxyphenylacetic acid glucuronide and sulfate than girls. Our findings are in line with those of Zamora et al., who observed that the median urinary concentrations of tyrosol, vanillic acid, and 4-hydroxyphenylacetic acid were at least 1.4-fold higher in men than women [31]. Similarly, Mumford et al., found higher values of enterodiol and enterolactone in females than males [34]. As sex hormones may be responsible for these differences [8,35], a limitation of the current study is that the follicular phase of the menstrual cycle was not considered during the urine collection to minimize bias related to the hormonal status of the participants.

#### 3.3.3. Urinary MPM and Dietary (Poly)Phenols

No differences were found between classes of urinary MPM and tertiles of total phenolic intake in the PCA (Appendix A). However, positive correlations were observed between urinary hydroxycoumarins (urolithins) and flavonoid intake and TPI (Figure 2). Additionally, positive correlations were observed between urinary lignans and intake of whole grains (R = 0.13, FDR-adjusted *p* = 0.007) and green-leaf vegetables (R = 0.13, FDR-adjusted *p* = 0.008). Urinary hydroxycinnamic acids also correlated with whole grains (R = 0.11, FDR-adjusted *p* = 0.015), green-leaf vegetables (R = 0.15, FDR-adjusted p = 0.002), and tomato or tomato-based products (R = 0.12, FDR-adjusted *p* = 0.011) (Appendix A). Urolithins are produced by gut microbiota through the metabolism of ellagitannins [11,36], whose main food sources are red fruits, nuts, and seeds [36], but in our study, urolithins were only positively correlated with nuts and seeds (R = 0.13, FDR-adjusted *p* = 0.014).

## 4. Conclusions

In conclusion, an HPLC-ESI-LTQ-Orbitrap-HRMS method was developed and fully validated to quantify urinary MPM in terms of linearity, sensitivity, recovery, accuracy, and precision. To our knowledge, this is the first time that several MPM have been identified and quantified in urine samples of an adolescent population using an HPLC-ESI-LTQ-Orbitrap-HRMS method on a large scale. Variations in MPM were observed between participants, which were associated with variability in dietary (poly)phenol intake and sex. Finally, some MPM were found to be potential dietary biomarkers of specific food groups, namely lignans for whole grains and urolithins for nuts. Further investigations are needed to explore the relationship between MPM and dietary sources of (poly)phenols.

## Figures and Tables

**Figure 1 antioxidants-11-01167-f001:**
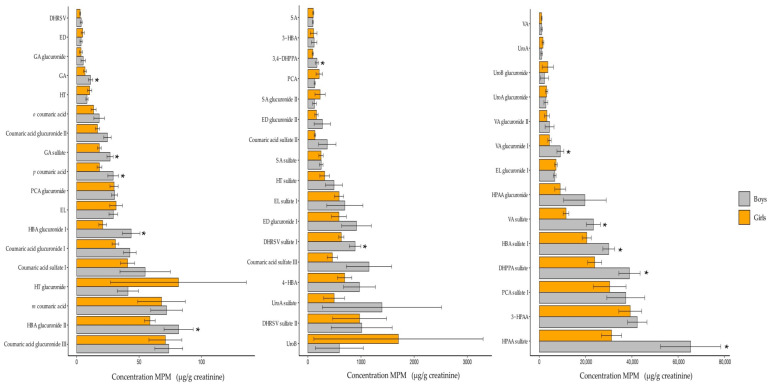
Urinary MPM of adolescents by gender. 3,4-DHPPA 3,4-dihydroxyphenylpropionic acid, 3-HPAA 3-hydroxyphenylacetic acid, 3-HBA 3-hydroxybenzoic acid, 3-HT 3-hydroxytyrosol, 3-HT-G 3-hydroxytyrosol glucuronide, 4-HBA 4-hydroxybenzoic acid, DHRSV dihydroresveratrol, ED enterodiol, EL enterolactone, GA gallic acid, PCA protocatechuic acid, SA syringic acid, Uro-A urolithin A, uro-B urilithin B, VA vanillic acid. Bar graphs are plotted as the mean (SEM). * *p*-values < 0.05 from *t*-test analysis.

**Figure 2 antioxidants-11-01167-f002:**
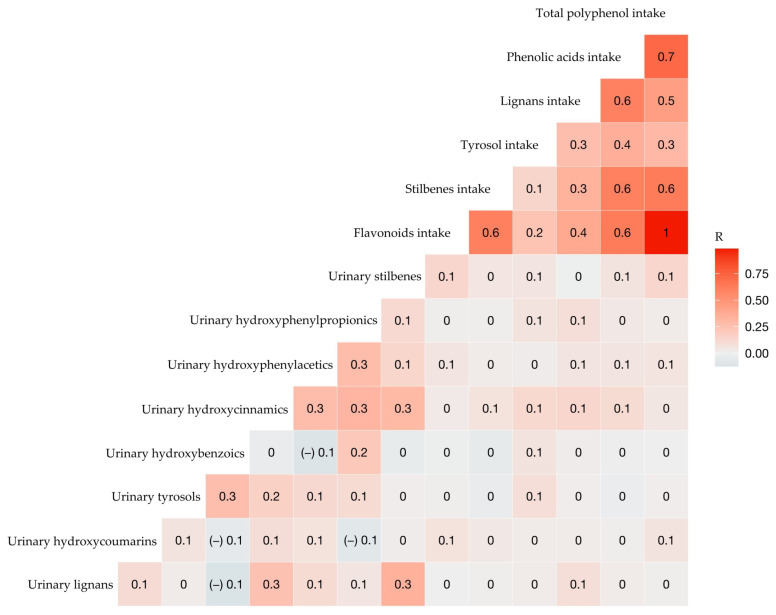
Heatmap of the Pearson correlation between subclasses of urinary MPM and energy-adjusted (poly)phenol intake in adolescents.

**Table 1 antioxidants-11-01167-t001:** Concentration levels of each phenolic standards for HPLC/ESI-LTQ-Orbitrap-HRMS method validation.

Phenolic Standards	Concentration Level (µg/L)		
	Low	Medium	High
Enterodiol	5	200	766
3′-Hydroxytyrosol-3′-glucuronide	5	200	766
3-Hydroxybenzoic acid	5	200	766
3-Hydroxytyrosol	5	200	766
4-Hydroxybenzoic acid	5	200	766
Enterolactone	5	200	766
*m*-coumaric acid	5	200	766
*p*-coumaric acid	5	200	766
Protocatechuic acid	5	200	766
*o*-coumaric acid	5	200	766
Syringic acid	5	200	766
Urolithin-B	5	200	766
Vanillic acid	5	200	766
Dihydroresveratrol	12.5	500	1915
Gallic acid	12.5	500	1915
Urolithin-A	12.5	500	1915
3,4-Dihydroxyphenylpropionic acid	25	100	3830
3′-Hydroxyphenylacetic acid	50	2000	7660

HPLC high performance liquid chromatography, ESI electrospray ionization, LTQ linear ion trap quadrupol, Orbitrap-HRMS.

**Table 2 antioxidants-11-01167-t002:** General characteristics of the participants.

	N	Mean (SD)	Median (IQR)
Age, years	601	12.0 (0.4)	12.0 (0.0)
Body mass, kg	601	50.8 (12.2)	48.5 (14.8)
Height, cm	601	155.2 (6.9)	155.2 (9.2)
BMI, kg/m^2^	601	20.9 (4.2)	20.1 (5.0)
BMI z-score	601	0.6 (1.0)	0.6 (1.4)
Energy and nutrients intake			
Energy, kcal/day	546	2498.8 (579.6)	2474.9 (828.6)
Carbohydrates, g/day	546	132.1 (47.3)	124.3 (63.0)
Fiber, g/day	546	29.6 (10.6)	28.1 (13.5)
Fat, g/da	546	112.0 (32.8)	109.4 (41.6)
Protein, g/day	546	119.5 (32.3)	117.8 (42.1)
Energy-adjusted (poly)phenol intake			
Total (poly)phenol intake, mg/day	546	683.5 (335.3)	639.8 (354.9)
Flavonoids, mg/day	546	533.9 (310.3)	480.8 (298.8)
Stilbenes, mg/day	546	0.2 (0.3)	0.1 (0.2)
Tyrosols, mg/day	546	21.3 (13.7)	17.8 (12.6)
Lignans, mg/day	546	3.7 (4.1)	2.5 (2.5)
Phenolic acids, mg/day	546	94.9 (50.4)	89.2 (51.7)

BMI: body mass index, IQR: interquartile range, SD standard deviation Values are given as means (SD) and medians (IQR).

**Table 3 antioxidants-11-01167-t003:** Quantification of urinary MPM by HPLC-ESI-LTQ-Orbitrap-HRMS.

Urinary MPM, µg/g Creatinine	<LOQ (*n*)	Mean *	SEM *	CV *
Lignans				
Enterodiol ^a^	136	4.5	0.9	1.0
Enterodiol glucuronide I (ED)	4	740.7	151.1	4.8
Enterodiol glucuronide II (ED)	3	209.2	67.8	5.1
Enterodiol sulfate (ED)	18	158.0	34.1	4.9
Enterolactone ^a^	179	30.6	3.2	1.7
Enterolactone glucuronide (EL)	3	6984.5	419.2	1.5
Enterolactone sulfate (EL)	19	639.3	168.8	6.3
Phenolic acids—Hydroxybenzoic acids				
Gallic acid ^a^	223	9.1	1.1	1.4
Gallic acid glucuronide (GA)	80	4.6	1.2	1.4
Gallic acid sulfate (GA)	87	22.8	1.5	1.3
3-Hydroxybenzoic acid ^a^	206	113.4	41.1	5.6
4-Hydroxybenzoic acid ^a^	1	824.5	157.8	4.6
Hydroxybenzoic acid glucuronide I (HBA)	229	33.4	4.2	1.3
Hydroxybenzoic acid glucuronide II (HBA)	13	69.5	6.0	1.8
Hydroxybenzoic acid sulfate (HBA)	0	25,034.4	1607.6	1.6
Protocatechuic acid ^a^	1	173.8	31.7	4.2
Protocatechuic acid glucuronide (PCA)	57	30.2	2.1	1.5
Protocatechuic acid sulfate I (PCA)	3	33,703.3	5368.2	3.8
Protocatechuic acid sulfate II (PCA)	0	228.0	41.8	3.6
Syringic acid ^a^	4	99.6	6.7	1.3
Syringic acid glucuronide I (SA)	0	297.6	26.8	2.0
Syringic acid glucuronide II (SA)	2	181.0	53.5	3.4
Syringic acid sulfate (SA)	32	249.9	26.9	1.8
Vanillic acid ^a^	0	1027.5	198.9	3.5
Vanillic acid glucuronide I (VA)	16	6847.5	857.4	2.5
Vanillic acid glucuronide II (VA)	2	3795.8	1038.3	4.7
Vanillic acid sulfate (VA)	1	17,227.2	1610.6	2.2
Phenolic acids—Hydroxycinnamic acids				
*m*-Coumaric acid ^a^	38	69.9	11.8	2.8
*o*-Coumaric acid ^a^	42	15.8	2.4	1.6
*p*-Coumaric acid ^a^	16	23.4	2.3	1.6
Coumaric acid glucuronide I	18	36.5	2.8	1.6
Coumaric acid glucuronide II	162	20.4	1.7	1.2
Coumaric acid glucuronide III	11	72.4	8.8	2.7
Coumaric acid sulfate I	39	46.8	9.2	2.6
Coumaric acid sulfate II	13	240.4	75.8	6.1
Coumaric acid sulfate III	5	788.7	208.6	5.3
Phenolic acids—Hydroxyphenylacetic acids				
3-Hydroxyphenylacetic acid ^a^	13	40,797.6	3248.4	1.8
Hydroxyphenylacetic acid glucuronide (3-HPAA)	122	13,860.5	4363.6	5.3
Hydroxyphenylacetic acid sulfate (3-HPAA)	22	45,815.5	6160.0	2.4
Phenolic acids—Hydroxyphenylpropanoic acids				
3,4-dihydroxyphenylpropionic acid ^a^	25	132.8	17.1	2.0
Dihydroxyphenylpropionic acid sulfate (3,4-DHPPA)	1	30,942.7	2700.1	2.0
Stilbenes				
Dihydroresveratrol ^a^	78	3.3	0.5	0.5
Dihydroresveratrol sulfate I (DHR)	4	753.5	57.8	1.8
Dihydroresveratrol sulfate II (DHR)	47	991.6	379.0	5.2
Other polyphenols—Hydroxycoumarins				
Urolithin A ^a^	57	1338.1	270.3	2.4
Urolithin A glucuronide (Uro A)	41	3030.2	482.1	2.7
Urolithin A sulfate (Uro A)	26	801.0	399.9	3.5
Urolithin B ^a^	86	1334.4	1067.1	4.4
Urolithin B glucuronide (Uro B)	63	3062.8	1565.1	6.1
Other polyphenols-Tyrosols				
3-Hydroxytyrosol ^a^	143	9.1	0.9	0.6
3′hydroxytyrosol-3′-glucuronide ^a^	71	62.4	28.9	7.5
Hydroxytyrosol sulfate (3-HT)	5	398.0	88.8	5.0

3,4-DHPPA 3,4-dihydroxyphenylpropionic acid, 3-HPAA 3-hydroxyphenylacetic acid, 3-HBA 3-hydroxybenzoic acid, 3-HT 3-hydroxytyrosol, 3-HT-G 3-hydroxytyrosol glucuronide, 4-HBA 4-hydroxybenzoic acid, DHRSV dihydroresveratrol, ED enterodiol, EL enterolactone, GA gallic acid, PCA protocatechuic acid, SA syringic acid, Uro A urolithin A, Uro B urilithin B, VA vanillic acid, LOQ limit of quantification, SEM mean standard error, CV coefficient of variance. When standards were not available, aglycone was used for quantification. The molecule used for the quantification is shown in brackets. ^a^ Commercial standards. * Data obtained from samples with microbial phenolic metabolites quantified by HPLC-ESI-LTQ-Orbitrap-HRMS. This table does not include data below the LOQ or non-detected compounds.

## Data Availability

There are restrictions on the availability of the data for the SI! Program study due to signed consent agreements around data sharing, which only allow access to external researcher for studies following project purposes. Requesters wishing to access the database used in this study can make a request to the Steering Committee (SC) chair: gsantos@fundacionshe.org, rodrigo.fernandez@cnic.es, juanmiguel.fernandez@cnic.es, RESTRUCH@clinic.cat, lamuela@ub.edu, bibanez@cnic.es, vfuster@cnic.es. For the present study, the database was requested from the SC on 24 February 2022.

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
