# Peer review of "Identification and Quantification of Urinary Microbial Phenolic Metabolites by HPLC-ESI-LTQ-Orbitrap-HRMS and Their Relationship with Dietary Polyphenols in Adolescents"

_antioxidants, 2022, doi:10.3390/antiox11061167_

Round 1
Reviewer 1 Report
This study is aimed to development and validation of the high-performance liquid chromatography/electrospray ionization-linear ion trap quadrupole-Orbitrap-high-resolution mass spectrometry (HPLC/ESI-LTQ-Orbitrap-HRMS) method to identify and quantify urinary MPM in adolescents and to explore the relationship of MPM with dietary (poly)phenols.
However, some changes are needed:
- Please see the suggested form of the abstract writing (without numbering and subtitles)
- Table 1 - give the concentration ascending or descending, also is confusing that the first high concentration is in the range of medium concentrations - so it would be much clearer if you would separate the table based on the phenolic standards (last column should be the first and 4 levels are needed)
- Table 2, weight is in kg and a more common term would be body mass
- why the N differs for the general anthropometric characteristics vs energy and nutrient intake?
- it is not explained what is IQR
- Figure 1: is in the Figs presented the mean with additional SD or SE?
- Figure S1: Does it present the mean ±SD or SE? I assume the unit on the x-axis should be μg L-1
- Figure S2: it is not Kelvin kcal it is kcal or Cal. Also, this figure should be shown in colors that differ significantly because it is not clearly visible in Figures S2 A, B and D whether individual dots are different colors or not.
- Text under figure S2 - the BMI is given in kg/m2 or "kg m-2", please do not mix two ways of writing measurement units
- Table S1: R2 can either be coefficient of determination or R-Squared is a statistical measure of fit that indicates how much variation of a dependent variable is explained by the independent variable(s) in a regression model, but it is certainly not a coefficient of regression, because this is something else in the model
- in the title of table s2 - missing a closed parenthesis
Sincerely,
Author Response
Response to Reviewer 1 Comments
Point 1: This study is aimed to development and validation of the high-performance liquid chromatography/electrospray ionization-linear ion trap quadrupole-Orbitrap-high-resolution mass spectrometry (HPLC/ESI-LTQ-Orbitrap-HRMS) method to identify and quantify urinary MPM in adolescents and to explore the relationship of MPM with dietary (poly)phenols. However, some changes are needed
Response 1: We appreciate greatly the helpful and detailed comments done by Reviewer#1 to our manuscript. Please refer below for detailed responses.
Point 2: Please see the suggested form of the abstract writing (without numbering and subtitles)
Response 2:
Following the reviewer’s suggestion, we have deleted the number and subtitles in the abstract (Page 1).
Point 3: Table 1 - give the concentration ascending or descending, also is confusing that the first high concentration is in the range of medium concentrations - so it would be much clearer if you would separate the table based on the phenolic standards (last column should be the first and 4 levels are needed).
Response 3:
Following the reviewer’s suggestion, we have changed Table 1 separating the table based on the phenolic standards (Page 4, line 147)
Point 4: Table 2, weight is in kg and a more common term would be body mass
- why the N differs for the general anthropometric characteristics vs energy and nutrient intake?
- it is not explained what is IQR
Response 4:
Following the reviewer’s suggestion, we have changed Table 2 changing “weight” by “body mass” expressed as kg. (Line 324)
We apologize because number of participants was not clear in the original version of the manuscript submitted. Out of 601 randomized participants with available urine samples selected in this cross-sectional analysis, 546 had also available information of food intake. This information has been considered in the revised version of the manuscript (Section 3.3.1. General characteristics of the study population).
IQR had been clarified in the table footnote in the original version of the manuscript (Line 325)
Point 5: Figure 1: is in the Figs presented the mean with additional SD or SE?
Response 5:
In Figure 1, bar graphs are plotted as the mean with an additional standard error of the mean (SEM). This information had been described in the table footnote in the original version of the manuscript but also in methods, in the section on data analysis. (Manuscript, Figure 1).
Point 6: Figure S1: Does it present the mean ± SD or SE? I assume the unit on the x-axis should be μg L-1
Response 6:.
In Figure S1, bar graphs are plotted as the mean with an additional standard error of the mean (SEM). This information had been described in the table footnote in the revised version of the manuscript but also in methods, in the section on data analysis. Additionally, the unit has been changed by µg/L. (Supplementary data, Figure S2, line 127).
Point 7: Figure S2: it is not Kelvin kcal it is kcal or Cal. Also, this figure should be shown in colors that differ significantly because it is not clearly visible in Figures S2 A, B, and D whether individual dots are different colors or not.
Response 7:
Following the reviewer’s suggestion, we have changed Figure S2 indicating energy intake as kcal and putting on colors but also indicating as asterisks, which is significant (Supplementary data, Figure S3, line 140).
Point 8: Text under figure S2 - the BMI is given in kg/m2 or "kg m-2", please do not mix two ways of writing measurement units
Response 8:
Following the reviewer’s suggestion, we have changed Figure S2 indicating BMI as kg/m2 (Supplementary data, Figure S3, line 140).
Point 9: Table S1: R2 can either be coefficient of determination or R-Squared is a statistical measure of fit that indicates how much variation of a dependent variable is explained by the independent variable(s) in a regression model, but it is certainly not a coefficient of regression, because this is something else in the model
Response 9:
Following the reviewer’s suggestion, we have considered “R2” like the coefficient of determination. (Supplementary data, Table S2, line 209).
Point 10: in the title of table s2 - missing a closed parenthesis
Response 10: Following the reviewer’s suggestion, we have added “)”, (Supplementary data, Table S3, line 232).
Reviewer 2 Report
I have read a manuscript entitled “Identification and quantification of urinary microbial phenolic metabolites by HPLC-ESI-LTQ-Orbitrap-HRMS and their relationship with dietary polyphenols in adolescents.” The authors have done a lot of work, but still I do not understand the key points.
Dear authors! According to the Introduction, the aim of your research was to develop and validate a high-performance liquid chromatography/electrospray ionization-linear ion trap quadrupole-Orbitrap-high-resolution mass spectrometry (HPLC/ESI-LTQ-Orbitrap-HRMS) method to identify and quantify urinary MPM in adolescents and to explore the relationship of MPM with dietary (poly)phenols. However, there is no development of the method in the manuscript. In the work devoted to the development of the method of analysis, there are no eluent selection conditions, no different chromatographic columns, no mass spectrometric analysis conditions, no temperature conditions, etc. Only a variation of the selection of eluent with formic acid and two injected volumes were indicated. Besides, the authors write that the data are not shown.
The authors validate the following compounds: 3-hydroxybenzoic acid, 3-hydroxytyrosol, 3′-hydroxytyrosol-3′-glucuronide, protocatechuic acid, 4-hydroxybenzoic acid, vanillic acid, syringic acid, enterodiol, enterolactone, urolithin-B, gallic acid, dihydroresveratrol, urolithin-A, 3,4-dihydroxyphenylpropionic acid, 3′-hydroxyphenylacetic acid, o-coumaric acid, m-coumaric acid, and p-coumaric acid. However, Table 3 lists compounds that have not been validated. What does it mean? Have some compounds been validated and others not? Also why compound µg/g creatinine is the unit in Table 3? According to line 115, the concentration of creatinine is only 1.1 h/L, while urea is 25 g/L.
It is also not clear from the Introduction why adolescents were chosen as volunteers. What are the criteria for selecting volunteers for this research study?
Thus, the authors should decide on the purpose of the study and reconsider the objectives and structure of the manuscript.
Author Response
Response to Reviewer 2 Comments
Point 1: I have read a manuscript entitled “Identification and quantification of urinary microbial phenolic metabolites by HPLC-ESI-LTQ-Orbitrap-HRMS and their relationship with dietary polyphenols in adolescents.” The authors have done a lot of work, but still I do not understand the key points.
Response 1: We appreciate greatly the helpful and detailed comments done by Reviewer#2 on our manuscript. The mainly key points of our manuscript are accordance with our objective described in the original manuscript: the development a quantitative method to determine the concentration of microbial phenolic metabolites (MPM) in urine samples of adolescents, and explore the relationship between these MPM with polyphenols dietary intake. Please refer below for detailed responses.
Point 2: Dear authors! According to the Introduction, the aim of your research was to develop and validate a high-performance liquid chromatography/electrospray ionization-linear ion trap quadrupole-Orbitrap-high-resolution mass spectrometry (HPLC/ESI-LTQ-Orbitrap-HRMS) method to identify and quantify urinary MPM in adolescents and to explore the relationship of MPM with dietary (poly)phenols. However, there is no development of the method in the manuscript. In the work devoted to the development of the method of analysis, there are no eluent selection conditions, no different chromatographic columns, no mass spectrometric analysis conditions, no temperature conditions, etc. Only a variation of the selection of eluent with formic acid and two injected volumes were indicated. Besides, the authors write that the data are not shown.
Response 2: We apologize because this was not clear in the original version of the manuscript submitted. The description of the validated method has been followed according to the criteria of the Association of Official Agricultural Chemists (AOAC) International in terms of linearity, limit of detection (LOD), limit of quantification (LOQ), recovery, intra- and inter-day accuracy and precision, and postpreparative stability. It has been described in the original version of the manuscript. However, for the development of this method, different tests were applied in terms of the cartridge of solid-phase extraction (SPE) selection, sample extraction solutions, phase-reverse chromatographic columns, mobile phase of liquid chromatogram, injection volume, and so on.
Following the reviewer’s suggestion, we have added complementary information in supplementary data, details about SPE cartridge selection, optimization of SPE with different solutions, as well as reverse-phase chromatographic columns tested to obtain better recoveries (Manuscript, section 3.1. Optimization of the HPLC/ESI-LTQ-Orbitrap-HRMS method, lines 252-265; and in Supplementary data, Analytical condition testing before validation HPLC/ESI-LTQ-Orbitrap-HRMS method, lines 50-104, Figure S1 and Table S1).
Regarding the spectrometric analysis conditions, they are the same for all components. Due to the characteristics of the mass spectrometer Orbitrap, individual conditions per each compound were not possible to set.
In addition to this, we have not described all the data of the tests done previously validation method because it would be tedious . Therefore, we have selected the most relevant information according to AOAC (Manuscript, section 3.1. Optimization of the HPLC/ESI-LTQ-Orbitrap-HRMS method, lines 252-265; and in Supplementary data, Analytical condition testing before validation HPLC/ESI-LTQ-Orbitrap-HRMS method, lines 50-104, Figure S1 and Table S1).
Point 3: The authors validate the following compounds: 3-hydroxybenzoic acid, 3-hydroxytyrosol, 3′-hydroxytyrosol-3′-glucuronide, protocatechuic acid, 4-hydroxybenzoic acid, vanillic acid, syringic acid, enterodiol, enterolactone, urolithin-B, gallic acid, dihydroresveratrol, urolithin-A, 3,4-dihydroxyphenylpropionic acid, 3′-hydroxyphenylacetic acid, o-coumaric acid, m-coumaric acid, and p-coumaric acid. However, Table 3 lists compounds that have not been validated. What does it mean? Have some compounds been validated and others not? Also why compound µg/g creatinine is the unit in Table 3? According to line 115, the concentration of creatinine is only 1.1 h/L, while urea is 25 g/L.
Response 3: We apologize because this was not clear in the original version of the manuscript submitted. According to point 2.5. “Validation of the HPLC/ESI-LTQ-Orbitrap-HRMS method, lines 138-200)” of the original manuscript, we described that our method was validated using the following commercial phenolic compound standards: 3-hydroxybenzoic acid, 3-hydroxytyrosol, 3′-hydroxytyrosol-3′-glucuronide, protocatechuic acid, 4-hydroxybenzoic acid, vanillic acid, syringic acid, enterodiol, enterolactone, urolithin-B, gallic acid, dihydroresveratrol, urolithin-A, 3,4-dihydroxyphenylpropionic acid, 3′-hydroxyphenylacetic acid, o-coumaric acid, m-coumaric acid, and p-coumaric acid. However, during the analysis of the 601 urine samples, we have semi-targeted identified phase II metabolites (glucuronides and sulfates) as we have mentioned in detail in the section “2.6.1 Targeted identification of MPM – lines 203-213”. For the quantification of these phase II metabolites, calibration curves of the phenolic compound’s standards were used, following the previously validated method. All details of the quantification of these MPM are described in section 2.62. “Quantification of MPM, lines 214-223” in the original manuscript. Thus, in Table 3 (line 345) we have reported MPM that included the concentration of the compounds with available commercial standard previously used in the method validation, as well as the concentration of phase II metabolites (glucuronides and sulfates) calculated using the curve calibration of their aglycone form (commercial standards). This information had also been described in the table footnote in the original version of the manuscript.
Point 4: Also why compound µg/g creatinine is the unit in Table 3? According to line 115, the concentration of creatinine is only 1.1 h/L, while urea is 25 g/L.
Response 4: MPM concentration was normalized by urinary creatinine concentrations in the 601 spot urine samples, which were determined using the Jaffé alkaline picrate method adapted to microtiter 96-well plates[1], and expressed finally as µg MPM/g creatinine.
Regarding the description in line 115, we have described the components from the synthetic urine used as blank. It contains calcium chloride (0.65 g/L), magnesium chloride (0.65 g/L), sodium chloride (4.6 g/L), sodium sulfate (2.3 g/L), sodium citrate (0.65 g/L), dihydrogen phosphate (2.8 g/L), potassium chloride (1.6 g/L), ammonium chloride (1.0 g/L), urea (25 g/L), and creatinine (1.1 g/L)[2] (Section 2.3. sample preparation and extraction of (poly)phenols). It is necessary to mention that synthetic urine was used in the calibration curves of commercial standards.
Point 5: It is also not clear from the Introduction why adolescents were chosen as volunteers. What are the criteria for selecting volunteers for this research study?
Response 5: We apologize because this was not clear in the original version of the manuscript submitted. In section 2.1. “Study design and sample selection”, we described the selection of 601 randomly chosen participants (53% girls) with available baseline urine samples for this cross-sectional study. The present study work was carried out as a cross-sectional analysis derived from the SI! (Salud Integral) Program for Secondary Schools trial in Spain, a cluster-randomized controlled intervention trial (NCT03504059) aiming to evaluate the impact of a lifestyle educational program on cardiometabolic health in adolescents. Details of the study design, recruitment procedures, and Commission on Ethics are available elsewhere[3]. A total of 1326 participants were recruited in the baseline of the Program for Secondary Schools trial1. From them, 1313 had urine samples, and for this analysis, we randomized selected 601 participants, equivalent to 45% of the cohort.
We have added the total of participants recruited in the SI! (Salud Integral) Program for Secondary Schools trial (lines 82-83), and the percentage of the participant randomized selected for the present cross-sectional study in the revised version of the manuscript to clarify this observation (lines 87-88).
Point 6: Thus, the authors should decide on the purpose of the study and reconsider the objectives and structure of the manuscript.
Response 6: We thank the reviewer for his/her comment. However, we considered that the original manuscript has been developed in accordance with our objective: “to develop and validate a high-performance liquid chromatography/electrospray ionization-linear ion trap quadrupole-Orbitrap-high-resolution mass spectrometry (HPLC/ESI-LTQ-Orbitrap-HRMS) method to identify and quantify urinary MPM in adolescents and to explore the relationship of MPM with dietary (poly)phenols”. Thus, we have presented first the validation of the HPLC/ESI-LTQ-Orbitrap-HRMS method following the criteria of the Association of Official Agricultural Chemists (AOAC) International in terms of linearity, the limit of detection (LOD), the limit of quantification (LOQ), recovery, intra- and inter-day accuracy and precision, and postpreparative stability[4]. Then, we have described the identification and quantification of the MPM in the 601 urine samples from adolescents (for this last using the developed validated method). Also, we described the procedure to estimate dietary polyphenols based on a semiquantitative food frequency questionnaire. Finally, we have presented the results about the relationship between MPM and dietary polyphenols in adolescents.
References:
[1] Medina-Remón, A.; Barrionuevo-González, A.; Zamora-Ros, R.; Andres-Lacueva, C.; Estruch, R.; Martínez-González, M.-Á.; Diez-Espino, J.; Lamuela-Raventos, R.M. Rapid Folin–Ciocalteu Method Using Microtiter 96-Well Plate Cartridges for Solid Phase Extraction to Assess Urinary Total Phenolic Compounds, as a Biomarker of Total Polyphenols Intake. Analytica Chimica Acta 2009, 634, 54–60, doi:10.1016/j.aca.2008.12.012.
[2] Roura, E.; Andrés-Lacueva, C.; Estruch, R.; Lamuela-Raventós, R.M. Total Polyphenol Intake Estimated by a Modified Folin–Ciocalteu Assay of Urine. Clinical Chemistry 2006, 52, 749–752, doi:10.1373/clinchem.2005.063628.
[3] Fernandez-Jimenez, R.; Santos-Beneit, G.; Tresserra-Rimbau, A.; Bodega, P.; de Miguel, M.; de Cos-Gandoy, A.; Rodríguez, C.; Carral, V.; Orrit, X.; Haro, D.; et al. Rationale and Design of the School-Based SI! Program to Face Obesity and Promote Health among Spanish Adolescents: A Cluster-Randomized Controlled Trial. American Heart Journal 2019, 215, 27–40, doi:10.1016/J.AHJ.2019.03.014.
[4] Lynch, J.; Horwitz, W.; Latimer, G.W. Appendix E: Laboratory Quality Assurance. Official Methods of Analysis of AOAC International. 2005.
Round 2
Reviewer 2 Report
The authors improved the manuscript according to the comments of the reviewers. I have no questions for the authors of the manuscript.
I recommend adding the HPLC method to the keywords.
I wish the authors success!